# Molecular Detection and Genetic Characterization of Vertically Transmitted Viruses in Ducks

**DOI:** 10.3390/ani14010006

**Published:** 2023-12-19

**Authors:** Xinrong Wang, Haidong Yu, Wenli Zhang, Lizhi Fu, Yue Wang

**Affiliations:** 1College of Veterinary Medicine, Southwest University, Chongqing 400715, China; 2Harbin Veterinary Research Institute, Chinese Academy of Agricultural Sciences, Harbin 150068, China; 3Chongqing Academy of Animal Science, Chongqing 408599, China; fulz@cqaa.cn

**Keywords:** duck hepatitis B virus, duck circovirus, duck hepatitis A virus 3, avian reoviruses, epidemiology

## Abstract

**Simple Summary:**

Vertically transmitted duck viruses are viruses that are transmitted from a female duck to its offspring when it is an egg, which seriously threatens production in the duck breeding industry. In this study, we evaluated the distribution and genetic variation in four vertically transmitted duck pathogens, including DHBV, DuCV, DHAV-3, and ARV. This study found that DHBV was the most prevalent virus, followed by DuCV, and then ARV and DHAV-3. The genetic analysis results showed that all the identified duck viruses here had complex qualities. These findings will improve our knowledge of the evolution of DuCV, DHAV-3, and ARV and help choose suitable strains for vaccination.

**Abstract:**

To investigate the distribution and genetic variation in four vertically transmitted duck pathogens, including duck hepatitis B virus (DHBV), duck circovirus (DuCV), duck hepatitis A virus 3 (DHAV-3), and avian reoviruses (ARV), we conducted an epidemiology study using PCR and RT-PCR assays on a duck population. We found that DHBV was the most prevalent virus (69.74%), followed by DuCV (39.48%), and then ARV (19.92%) and DHAV-3 (8.49%). Among the 271 duck samples, two, three or four viruses were detected in the same samples, indicating that the coinfection of vertical transmission agents is common in ducks. The genetic analysis results showed that all four identified DuCV strains belonged to genotype 1, the DHAV-3 strain was closely clustered with previously identified strains from China, and the ARV stain was clustered under genotype 1. These indicate that different viral strains are circulating among the ducks. Our findings will improve the knowledge of the evolution of DuCV, DHAV-3, and ARV, and help choose suitable strains for vaccination.

## 1. Introduction

Duck viral diseases pose a serious threat to the health and production performance of duck flocks and are prevalent worldwide. The highly pathogenic avian influenza, Newcastle disease virus, duck viral enteritis, and duck viral hepatitis are among the main culprits causing significant economic losses to duck flocks. Numerous studies have been conducted to address these diseases [1]. However, there are also several duck vertical transmission viral diseases that are often overlooked, including duck hepatitis B virus (DHBV), duck circovirus (DuCV), duck hepatitis A virus (DHAV), and avian reoviruses (ARV). These diseases can pose health risks to the duck industry, leading to problems such as duckling deformities, increased mortality rates, and poor growth and development [2,3,4,5,6].

DHBV, a member of the *Avihepadnavirus* genus in the *Hepadnaviridae* family, is a small DNA virus with a diameter of 40–45 nm. The DHBV model is widely utilized as an animal infection model for studying human HBV infections [7]. The previous reports have shown that ducks that are persistently infected with DHBV have reduced egg hatching rates during breeding and their ducklings have limited growth [8,9]. Although DHBV infections usually do not show serious symptoms or liver damage in ducks, a virus infection can affect the glucose metabolism by disrupting the glucose tolerance in the liver and other organs [10,11].

DuCV, a non-enveloped virus, belongs to the *Circovirus* genus of the *Circovirus* family and has a diameter of approximately 15–16 nm. Its viral genome is a single-stranded circular DNA of around 2.0 kilobases (kb) and contains two major open reading frames (ORFs) that encode the Rep gene and Cap proteins [12,13]. DuCVs are currently classified into two genotypes, DuCV-1 and DuCV-2, based on the complete genome and Cap gene sequence [14,15,16]. The prevalence of DuCV has been observed in various duck breeds, including Cherry Valley ducks, Pekin ducks, Muscovy ducks, mule ducks, and wild ducks [17,18,19,20,21,22,23]. The published data have demonstrated that a DuCV infection primarily affects the immune system and leads to immunosuppression, making the ducks more susceptible to other infections.

DHAV, a non-enveloped, single positive-stranded RNA virus, belongs to the *Avihepatovirus* genus in the *Picornaviridae* family [24,25]. The genome of DHAV contains an open reading frame (ORF) that encodes an inactive precursor protein. This precursor protein is subsequently cleaved into multiple viral proteins, including the structural protein P1 region (capsid proteins VP0, VP1, and VP3) and the non-structural proteins P2 region (2A, 2B, and 2C proteins) and P3 region (3A, 3B, 3C, and 3D proteins) [26,27,28,29,30,31]. DHAV is classified into three serotypes, DHAV-1, DHAV-2, and DHAV-3, based on the VP1 gene sequence. Currently, the most prevalent serotypes in China are DHAV-1 and DHAV-3 [32,33,34,35,36]. DHAV-1 is the classical serotype found worldwide. In China, the use of an officially approved live-attenuated vaccine for breeder ducks since 2013 has effectively controlled DHAV-1 infections. However, a high proportion of DHAV infections caused by DHAV-3 has been observed in China, which can cause liver damage, immune suppression, and death in some duck breeds [35,37,38].

ARV, a non-enveloped virus with a particle size from 70 to 80 nm, belongs to the *Orthoreovirus* genus of the *Reoviridae* family [39,40,41]. The genomes of ARV contain ten double-stranded RNA segments separated into three size classes, large (L1, L2, and L3), medium (M1, M2, and M3), and small (S1, S2, S3, and S4) according to their electrophoretic mobility, which encode twelve proteins, large (λA, λB, and λC), medium (μA, μB, and μC), and small (σC, P10, P17, σA, σB, and σNS) [42,43,44,45]. The σC is the most variable protein of ARV, which contains specific epitopes and is used as a genetic marker [46,47,48,49]. The phylogenetic analysis of the σC gene has showed that ARVs are divided into six genotypic clusters [50,51]. The ARVs are divergent in their pathogenicity and can infect a wide variety of avian species [51].

Despite the previous reports on these viruses, the epidemiological characteristics and co-infection rate of each virus in ducks have not been reported. Here, to understand the epidemiology and distribution of DHBV, DuCV, DHAV-3, and ARV in ducks, a surveillance study was conducted involving the PCR- and RT-PCR-based screening of anal swab samples collected from a duck population. To explore the genetic diversity of these detected viruses, the nucleotide sequences were analyzed and compared with some previously reported strains.

## 2. Materials and Methods

### 2.1. Specimen Collection

Between September 2020 and January 2022, a total of 271 fresh duck anal swabs were collected. The swabs were from two breeds of ducks from the same duck factory, including 152 Jinding duck swabs and 119 Shaoxing duck swabs. The fresh swab samples were stored in RNase-free containers at −80 °C.

### 2.2. Viral DNA/RNA Extraction and Reverse Transcription

Viral DNA/RNA was extracted from swabs following the procedure of a viral genome DNA/RNA extraction kit (Tiangen Biotech, Beijing, China). To determine the presence or absence of two RNA viruses, reverse transcription (RT) was initially performed to obtain cDNAs using a reverse transcription kit (TakaRa, Dalian, China).

### 2.3. PCR

To detect DNA viruses (DuCV and DHBV), viral DNA was extracted and used as the template. For RNA viruses (DHAV-1, DHAV-3, and ARV), viral RNA was extracted and used as the template to obtain cDNA using a reverse transcription kit. Following this, PCR was performed using the primers listed in Table 1. The positive PCR products were then recovered and ligated to a pMD18T vector for sequencing. To conduct a more detailed analysis of the molecular evolvement of these viruses, the full-length sequences of DuCV, the DHAV-3 VP1 gene fragment, and the ARV σC gene fragment were amplified using the primers provided in Table 2 and sequenced with some positive samples.

### 2.4. DNA Sequences and Phylogenetic Analysis

BLASTN was used to compare the nucleotide sequences with those available in the NCBI nucleotide database. MegAlign was used to compare the sequence homologies. Phylogenetic trees of three viruses were constructed using the neighbor joining and the Kimura 2-parameter model with the bootstrap analysis (1000 replicates) method in MEGA7.0. MegAlign 7.1.0 software was used to compare and analyze some amino acid sequences. Subsequently, ProtParam tool on the ExPASy website (Available from: http://www.expasy.org, accessed on 6 March 2023) was used to analyze the physicochemical properties of the strains obtained in this study and those previously popular strains (only marked with different amino acid sites), including the theoretical pI, instability index, aliphatic index, and grand average of hydropathicity (GRAVY).

## 3. Results

### 3.1. Prevalence of DHBV, DuCV, DHAV-3, and ARV in Ducks

In this study, a total of 271 anal swabs collected from Jinding and Shaoxing ducklings were analyzed using PCR and RT-PCR techniques to detect the presence of DHBV, DuCV, DHAV-3, and ARV (Figure 1). The positive samples were randomly selected and further confirmed by sequencing before conducting large-scale screening. Out of the total samples tested, 94.10% (255/271) were positive for at least one of the target viruses. The positive rates for DuCV, DHAV-3, DHBV, and ARV were 39.48%, 8.49%, 69.74%, and 19.92%, respectively (Table 3).

### 3.2. Coinfection of DHBV, DuCV, DHAV-3, and ARV

Furthermore, we conducted the analysis of the co-infection rates of the four viruses in ducks. Among the ducks that tested positive for at least one target virus, a co-infection rate of 38.04% (97/255) was observed. In addition, co-infections with three or even four viruses were also detected in the ducks (Figure 2). Out of the 255 ducks that were infected with at least one target virus, 22.75% (58/255) were found to be co-infected with DuCV and DHBV, 10.98% (28/255) were found to be co-infected with DuCV and ARV, 10.20% (26/255) were found to be co-infected with ARV and DHBV, and 1.96% (5/255) were found to be co-infected with DuCV, DHAV-3, and DHBV (Table 4).

### 3.3. Sequencing and Phylogenetic Analysis

To better understand the prevalent strains in the ducks, some positive samples were randomly selected for sequencing. Thus, no significant sequence variation was observed between the DHBV samples that tested positive in this study and the reference DHBV strains from GenBank. Therefore, further sequence analysis for DHBV was not conducted in the subsequent experiment.

The whole-genome sequences of four DuCV strains were obtained in this study (GenBank accession number: OQ658500-OQ658503). A phylogenetic tree of DuCVs was constructed by using the whole-genome sequences of these four newly DuCVs strains and forty-two reference strains from GenBank. The results showed that the four DuCV strains were clustered under the DuCV-1 genotype evolutionary branch with reference strains from China, the United States, Korea, and Brazil (Figure 3a). The nucleotide identity between them ranged from 82.4% to 99.8% (Table 5). Further sequence analysis showed that these four newly identified DuCV strains had 99.8% or higher nucleotide sequence similarity to the reference strain GD/ZQ/144 (ON227548) from Guangdong province in China. Amino acid sequence analysis revealed that the Cap proteins from these four DuCV-1 strain were identical (Table 6). However, several amino acid residues in the Cap protein of the four strains differed significantly from those in the reference strains (Figure 3b).

Five DHAV-3 positive samples were tested for VP1 gene amplification, and the sequence analysis revealed that these samples had identical sequences. The only GenBank accession number available for these samples is OQ658504. A phylogenetic tree was constructed based on the VP1 gene from the DHAV-3 strain identified here, along with 48 reference strains. The analysis results showed that the DHAV-3 strain identified here was closely clustered with the strains from Heihongjiang, Sichuan, and Shandong provinces in China, and this stain shared the most of its nucleotide identity, 99.72%, with one of the DHAV-3 strains from Heilongjiang province (Figure 4a). The homology in the VP1 gene sequence between the DHAV-3 strain here and the reference strains ranged from 66.9% to 100% (Table 7). Subsequent analysis revealed variations in multiple amino acid residues (Figure 4b), which aligned with the patterns observed in recent epidemic strains.

One σC gene was obtained from a positive ARV sample (GenBank accession number: OR046324) and used to conduct phylogenetic classification. The analysis was based on the σC gene alignment of the identified ARV strain and 49 reference strains. The results showed that the ARV strain detected in this study was clustered with the strains from China, Taiwan, the United States, Korea, and Brazil. The genetic evolution of the ARV σC gene showed that the identified ARV strain was distributed in branch I, which contains the largest number of reference strains in genotype I (Figure 5a). The nucleotide sequence homology of the σC gene showed that the ARV strain obtained here was the most homologous with other genotypes I strains, ranging from 75.7% to 98.3%. This was followed by genotype II, with a range from 54.2% to 60.5%; genotype III, with a range from 56.9% to 57.8%; genotype IV, with a range from 53.4% to 54.2%; genotype V, with a range from 51.1% to 54.0%; and genotype VI, with a range from 49.5% to 49.7% (Table 8). The nucleotide homologies of the ARV strain here with the strains from Canada (L39002), the United States (EF122836), Taiwan (AF204947), Brazil (DQ868789), and India (EU681254) were 98.2%, 98.8%, 98.0%, 99.5%, and 98.9%, respectively. These findings indicate that the ARV strain is widely distributed worldwide and has a significant global impact. The results OF amino acid sequence analysis showed that the observed changes in these amino acid residues are likely associated with the virus’s prevalence and spread, taking into account the year and region information (Figure 5b). These changes are consistent with those observed in recent epidemic strains.

In this study, the physicochemical properties of the obtained sequences were determined. The instability coefficients of these proteins were analyzed, taking into account the influence of different amino acid sites on their physicochemical properties (Table 9). The results showed that the instability coefficients of these proteins were significantly different, indicating variations in their structural stability. The stability of the viral proteins is crucial for their proper folding, assembly, and function. Changes in the physicochemical properties of viral proteins can affect their stability, potentially leading to alterations in viral replication, infectivity, and host interactions. Therefore, the observed differences in the instability coefficients of these proteins may contribute to the variations in the prevalence of different virus strains.

## 4. Discussion

China holds the distinction of being the largest producer and consumer of waterfowl globally. In fact, China’s duck production accounts for over 80% of the total worldwide quantity, making duck the third most consumed meat in the country, following pork and chicken [52]. As the duck breeding industry continues to intensify, the incidence of viral infectious diseases in ducks is also on the rise. In this study, we gathered evidence supporting the presence of four vertically transmitted viruses in duck populations, including DuCV, DHAV-3, DHBV, and ARV. We also observed the co-infections of these four viruses and evaluated the genotypes of DuCV, DHAV-3, and ARV. Additionally, the amino acid sequence and physicochemical properties of the proteins were analyzed.

DuCV, a novel virus, has garnered significant attention from the veterinary industry and has been reported in several countries, including Germany, Hungary, Taiwan in China, South Korea, and more [20,53,54,55]. Currently, DuCV is widely prevalent among the domestic duck populations [17,56]. DuCV infections in domestic ducks are typically subclinical and primarily affect the host’s immune system, leading to various secondary infections [14,53,57,58,59,60]. The high prevalence of DcCV has been reported in asymptomatic duck populations in China, with ducks of various age groups being susceptible to the virus. The DuCV-positive ducks exhibited a higher rate of infection by DHV-I, *Riemerella anatipestifer*, and *E. coli* compared to that of the DuCV-negative ducks [61,62]. In this study, we have identified four DuCV strains in the duck populations. Through sequence analysis, we determined that these viruses belong to genotype 1 and are closely clustered with the strains from Shandong, Hebei, and Guangdong provinces in China. Among the two genotypes of the virus, DuCV-1 and DuCV-2, there have been more reported cases of DuCV-1 in countries such as Germany, Hungary, the United States, China, South Korea, and Poland. However, phylogenetic analysis indicates that there is a significant variation in the genotype of DuCV in Mainland China, where both genotype 1 and genotype 2 are widely distributed and prevalent [60,63]. And previous epidemiological investigations have shown DuCV-1 to be more widespread and more pathogenic than DuCV-2 is [60]. Moreover, DuCV has been linked to emerging diseases, such as “beak atrophy and dwarfism syndrome (BADS)” and primary sclerosing cholangitis, which can cause severe clinical symptoms [12,18]. It is worth noting that gene recombination has consistently played a crucial role in the virus’ evolution. Several recombination events have been observed in circoviruses, such as recombination between porcine circovirus 2, porcine circovirus 3, goose circovirus, and DuCV [64,65,66,67,68]. However, our study did not yield sufficient evidence of recombination among the four DuCV isolates sequenced in this study. This limited evidence could be attributed to the limited number of DuCV sequences included in the analysis. Despite this, our findings indicate that these sequences align with significant changes in the amino acid sequence of the Cap protein observed in recent years. Additionally, our analysis based on the regional and yearly data suggest that 2020 may represent a crucial time period for significant changes in multiple amino acid sites, as depicted in Figure 3b. The pathobiological characteristics of DuCV-1 encompass systemic infection, persistent infection, as well as horizontal and vertical transmission [60]. Regrettably, despite advancements in the vaccine research, the absence of a robust culturing system has hampered the development of effective prevention methods and satisfactory antiviral drugs for DuCV. This limitation has posed challenges in both DuCV vaccine development and antiviral research [69,70].

DHAV is responsible for a severe and fast-spreading viral infections among young ducklings, characterized by high fatality rates reaching 95% and posing a significant economic risk to the duck industry [71]. On the other hand, adult ducks typically exhibit resistance to DHAV and do not show apparent clinical symptoms upon infection. However, the infected adult ducks can still shed the virus, serving as a potential source of infection for others [6]. However, a recent study has reported that a DHAV-1 infection can potentially trigger egg drop syndrome [72,73]. Furthermore, epidemiological investigations have indicated that following the widespread use of the officially approved DHAV-1 live vaccine in 2013, an increased incidence of DHAV infections caused by DHAV-3 was observed. In fact, the DHAV-3 strain has emerged as the predominant viral type in China [35,37,38,74,75]. In the present investigation, there were no instances of DHAV-1 detection, and therefore, the focus was solely on the genetic evolution analysis of the identified DHAV-3 strain. The identified DHAV-3 strain exhibited a significant relationship with the Sichuan and Shandong strains (MN912703, KX523286, and KX523289) from China, as they formed a distinct cluster within the evolutionary branch associated with the DHAV-3 genotype. Our analysis determined that this particular strain exhibited a close genetic relationship with the MN164467.1 strain, which was isolated from Heilongjiang, China. The nucleotide homology between them reached 99.72%. Notably, our findings indicated that the sequence displayed notable alterations in the amino acid sequence of the VP1 protein, reflecting the significant changes that have occurred in recent years. Despite these modifications, our analysis did not reveal any significant impact on the physicochemical properties of the proteins. However, further investigations are required to assess whether these alterations influence viral adsorption and the other related functions.

The σC protein of ARV encoded by the third open reading frame of the S1 genome fragment is a minor component of the virion coat and plays a crucial role in cell attachment and the induction of neutralizing antibodies [76]. Moreover, the ARV σC protein has a variety of functions, such as inducing apoptosis, an anti-tumor effect, and immune enhancement [77]. Notably, mutations in the σC protein are responsible for segment rearrangement, leading to genetic diversity within the reovirus population [72,76]. Mutations in the amino acid sites of this protein can lead to changes in pathogen infectivity, resulting in antigenic variation. Other studies have shown that when the amino acid sequence difference between the vaccine strain and the circulating strain is equal to or greater than 5%, the cross-protection effect of the vaccine is weakened [77]. The nucleotide and amino acid homologies between the σC sequence obtained in this study and the σC sequence of the S1133 vaccine strain were 98.8% and 97.9%, respectively. The amino acid sequence encoded by the major protective antigen σC gene of the current commercial vaccine strain S1133 is highly homologous with that of the ARV strain identified in this study, so we speculate that the existing vaccine can produce a protective effect against this variant strain. Other studies have shown significant genetic changes in certain regions of the ARVs genome and the emergence of ARVs genetic variation prior to 1990 [78]. Consequently, it is crucial to promptly comprehend the epidemic situation and pathogenic characteristics of ARV, and effectively manage the outbreak and prevalence of the disease by enhancing the prevention and control measures, bolstering the research and development of novel vaccines, and implementing a series of measures. 

The DHAV-3 VP1 and ARV σC sequences obtained in this study were closely related to those isolated and identified in Heilongjiang province. We hypothesize that virus transmission may have occurred due to trade within the province. On the other hand, the DuCV strains were closely clustered with the strains from Shandong, Hebei, and Guangdong. The similarity of DuCV strains in these areas may be related to the migration of wild ducks. In addition, these sequences are consistent with the changes in the amino acid sequences of major immunogenic proteins of epidemic strains in recent years. Furthermore, the amino acid sequences of the major immunogenic proteins of these epidemic strains have shown changes in recent years. This suggests ongoing viral evolution and adaptation, which may have implications for the effectiveness of existing vaccines and the development of new preventive measures. In order to gain a more comprehensive understanding of the vertical transmission of related viruses in ducks in mainland China, enhance biosafety breeding, and reinforce the prevention and control programs, it is necessary to broaden the geographical scope of the survey and contribute to the overall understanding and management of viral infectious diseases in the ducks in mainland China.

## 5. Conclusions

This study revealed the prevalence of DuCV, DHAV-3, DHBV, and ARV in ducks. It also demonstrates the coexistence of these viruses in healthy adult ducks. The detection of DuCV, DHAV-3, DHBV, and ARV in healthy adult ducks suggests a high likelihood of vertical transmission, leading to infections among ducklings. Hence, further research is required to gain a deeper understanding of these viruses’ role in ducks and their potential health implications.

## Figures and Tables

**Figure 1 animals-14-00006-f001:**
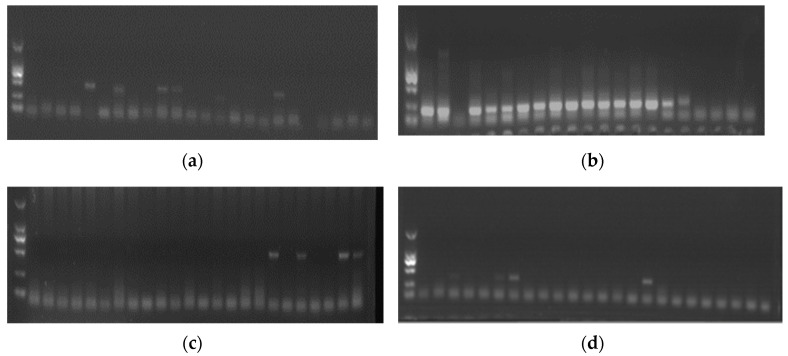
Part of PCR or RT-PCR results of DuCV (**a**), DHBV (**b**), DHAV-3 (**c**), and ARV (**d**).

**Figure 2 animals-14-00006-f002:**
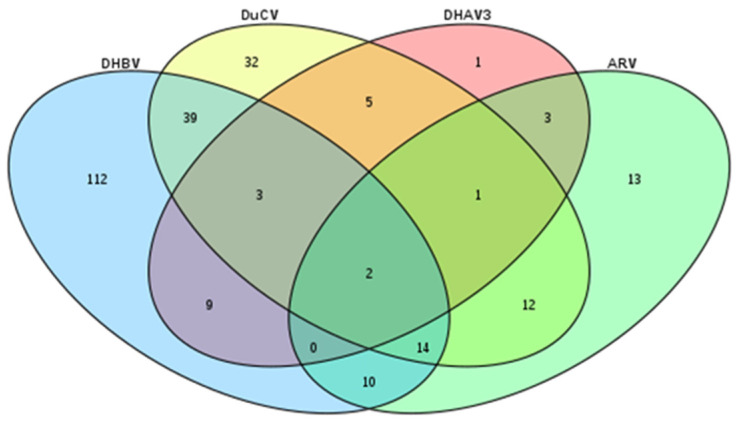
Coinfection rates of DHBV, DuCV, DHAV-3, and ARV.

**Figure 3 animals-14-00006-f003:**
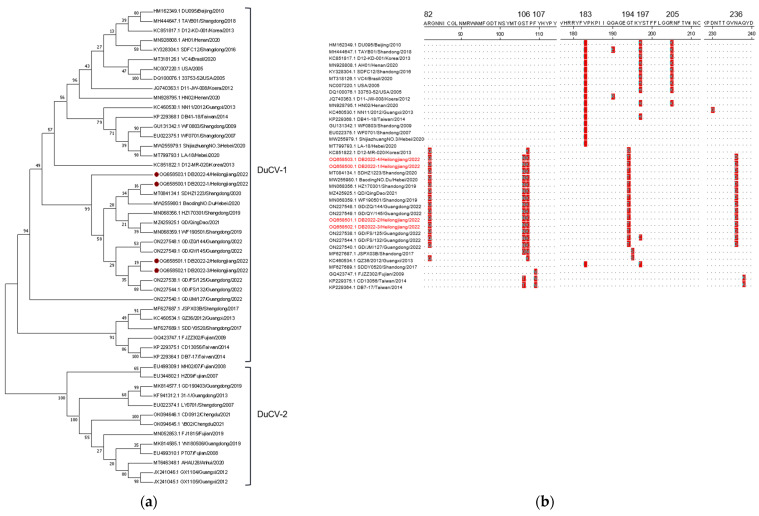
Sequence analysis of DuCV. (**a**): Phylogenetic tree of DuCV. Strains detected in this study are marked with “⦁”. (**b**): Amino acid sequence analysis of DuCV-Cap. Strains detected in this study are marked with red color.

**Figure 4 animals-14-00006-f004:**
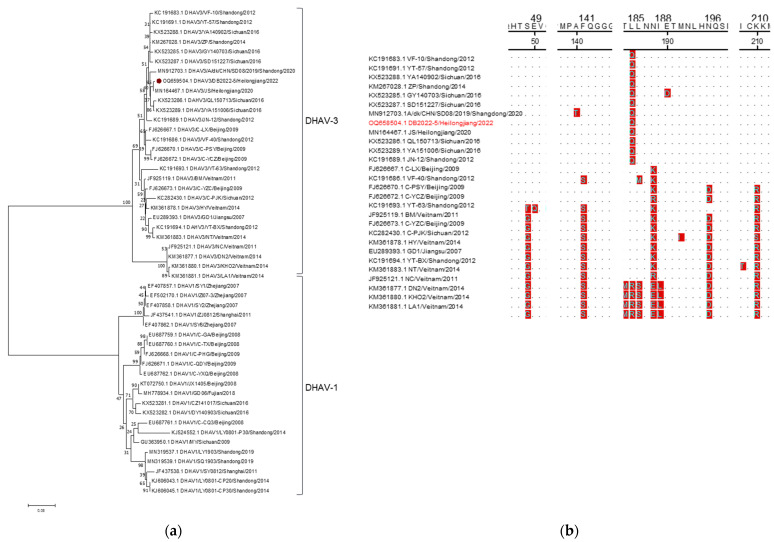
Sequence analysis of DHAV-VP1. (**a**): Phylogenetic tree of DHAV. (**b**): Amino acid sequence analysis of DHAV3-VP1. The DHAV strain identified in this study are marked with “⦁”. The strains detected in this study are marked in red.

**Figure 5 animals-14-00006-f005:**
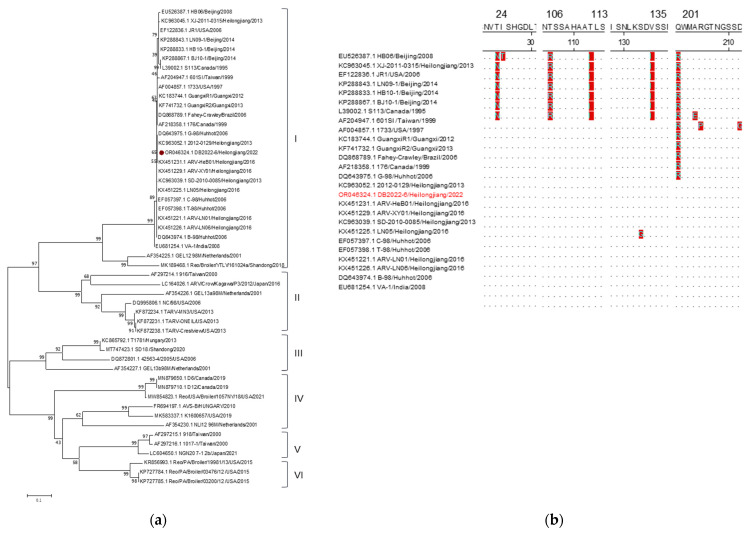
Sequence analysis of ARV. (**a**): Phylogenetic tree of ARV. The ARV strain identified in this study are marked with “⦁”. (**b**) B: Amino acid sequence analysis of ARV-σC. The strain detected in this study is marked in red.

**Table 1 animals-14-00006-t001:** Primers sequences used for detection.

Virus	Forward Primer (5′-3′)	Reverse Primer (5′-3′)	Fragment Size (bp)
DuCV	gcacgctcgacaattgcaagt	gccacgcccaaagattacataag	338
DHBV	gggctaggagattgctttg	ggttcgagtccacgaggtt	217
DHAV-1	agcttaaggcccggtgccccg	ggtagggtagggaatagtaaagt	399
DHAV-3	aacccctttgatccacactg	gataaggcatccacaccatc	544
ARV	taatttagacggtttgagga	cgttgagaacagaagtaggg	324

**Table 2 animals-14-00006-t002:** Primers sequences used for amplification.

Virus	Forward Primer (5′-3′)	Reverse Primer (5′-3′)	Fragment Size (bp)
DuCV	accggcgcttgtactccgtactcc	aataatataacggcgcttgtgcggt	1995
DHAV-3-VP1	ggtgattccaatcagcttggcga	ttcaatttctagatggagctcaaag	720
ARV-σC	atggcgggtctcaatccatcgca	ttaggtgtcgatgccggtacgcacg	981

**Table 3 animals-14-00006-t003:** The prevalence of DHBV, DuCV, DHAV-3, and ARV.

Duck Breed	Number of Samples	Positive Sample (%)			
		DHBV	DuCV	DHAV-3	ARV
Jinding	152	63.15 (96/152)	53.95 (82/152)	8.55 (13/152)	18.42 (28/152)
Shaoxing	119	78.15 (93/119)	21.01 (25/119)	8.40 (10/119)	21.85 (26/119)
Overall	271	69.74 (189/271)	39.48 (107/271)	8.49 (23/271)	19.92 (54/271)

**Table 4 animals-14-00006-t004:** Coinfection rates of DuCV, DHAV-3, DHBV, and ARV.

Virus	Coinfection Rate (%)	Virus	Coinfection Rate (%)
DHBV and DuCV	22.75	DHBV and ARV	10.20
DHBV and DHAV-3	5.49	DHBV, DHAV-3 and ARV	0.78
DuCV and ARV	10.98	DHBV, DuCV and DHAV-3	1.96
DuCV and DHAV-3	3.92	DuCV, DHAV-3 and ARV	0.78
DHAV-3 and ARV	1.96	DHBV, DuCV, DHAV-3 and ARV	0.78

**Table 5 animals-14-00006-t005:** Homology comparison of the whole genome of the identified DuCV with reference strains.

No	Strain	Year	Region	Accession No.	Cap Gene Sequence Identity (%)		Rep Gene Sequence Identity (%)		Full Genome Identity (%)
					nt	aa	nt	aa	nt
1	DU095	2010	Beijing/CN	HM162349.1	94.1%	96.5%	98.2%	97.6%	96.2%
2	TAYB01	2018	Shandong/CN	MH444647.1	93.9%	96.5%	98.4%	96.9%	96.4%
3	D12-KD-001	2013	Korea	KC851817.1	94.4%	96.9%	98.2%	98.0%	96.4%
4	AH01	2020	Henan/CN	MN928808.1	94.4%	96.5%	99.1%	97.6%	97.3%
5	SDFC12	2016	Shandong/CN	KY328304.1	94.2%	96.5%	98.5%	97.3%	97.1%
6	VC4	2020	Brazil	MT318126.1	93.5%	96.5%	98.2%	97.3%	96.4%
7	-	2005	USA	NC007220.1	93.5%	96.5%	98.1%	96.9%	96.2%
8	33753-52	2005	USA	DQ100076.1	93.5%	96.5%	98.1%	96.9%	96.2%
9	D11-JW-008	2012	Korea	JQ740363.1	94.7%	96.9%	98.2%	97.6%	97.0%
10	HN02	2020	Henan/CN	MN928795.1	94.4%	96.5%	99.2%	99.0%	97.4%
11	NN11/2012	2013	Guangxi/CN	KC460530.1	96.5%	96.1%	99.0%	99.0%	98.0%
12	DB41-18	2014	Taiwan/CN	KP229368.1	96.4%	97.3%	98.3%	98.0%	97.4%
13	WF0803	2009	Shandong/CN	GU131342.1	97.2%	97.7%	99.2%	99.0%	97.8%
14	WF0701	2007	Shandong/CN	EU022375.1	96.9%	97.3%	98.9%	99.0%	97.6%
15	ShijiazhuangNo. 3	2020	Hebei/CN	MW255979.1	97.2%	97.3%	98.8%	98.6%	97.9%
16	LA-18	2020	Hebei/CN	MT799793.1	96.6%	97.3%	98.5%	98.6%	97.6%
17	D12-MR-020	2013	Korea	KC851822.1	97.3%	98.8%	98.8%	99.0%	98.0%
18	SDHZ1223	2020	Shandong/CN	MT084134.1	99.9%	100.0%	99.9%	99.7%	99.7%
19	BaodingNO.Du	2020	Hebei/CN	MW255980.1	99.7%	100.0%	99.7%	99.7%	99.7%
20	HZ170301	2019	Shandong/CN	MN068356.1	99.9%	100.0%	99.8%	99.7%	99.8%
21	QD	2021	QingDao/CN	MZ425925.1	99.6%	99.6%	99.3%	98.6%	99.5%
22	WF190501	2019	Shandong/CN	MN068359.1	99.9%	100.0%	99.3%	99.0%	99.4%
23	GD/ZQ/144	2022	Guangdong/CN	ON227548.1	100.0%	100.0%	99.9%	99.7%	99.8%
24	GD/QY/145	2022	Guangdong/CN	ON227549.1	100.0%	100.0%	99.5%	99.3%	99.7%
25	GD/FS/125	2022	Guangdong/CN	ON227538.1	99.6%	99.6%	99.2%	98.3%	99.4%
26	GD/FS/132	2022	Guangdong/CN	ON227544.1	99.5%	98.8%	99.5%	98.6%	99.6%
27	GD/JM/127	2022	Guangdong/CN	ON227540.1	99.4%	100.0%	97.0%	95.9%	97.6%
28	JSPX03B	2017	Shandong/CN	MF627687.1	92.2%	97.3%	97.6%	97.6%	95.5%
29	QZ36/2012	2013	Guangxi/CN	KC460534.1	91.0%	97.7%	96.7%	95.9%	94.3%
30	SDDY0520	2017	Shandong/CN	MF627689.1	93.2%	96.9%	96.8%	96.9%	95.1%
31	FJZZ302	2009	Fujian/CN	GQ423747.1	92.2%	96.9%	97.8%	98.0%	95.2%
32	CD13056	2014	Taiwan/CN	KP229375.1	90.2%	96.5%	97.4%	98.0%	94.3%
33	DB7-17	2014	Taiwan/CN	KP229364.1	89.5%	95.7%	97.4%	98.0%	94.0%
34	MH02/07	2008	Fujian/CN	EU499309.1	79.1%	89.1%	92.4%	85.7%	84.8%
35	HZ09	2007	Fujian/CN	EU344802.1	78.9%	89.1%	92.3%	85.7%	85.2%
36	GD190403	2019	Guangdong/CN	MK814577.1	78.9%	88.4%	88.2%	82.9%	82.9%
37	31-1	2013	Guangdong/CN	KF941312.1	79.1%	88.4%	88.2%	82.9%	82.9%
38	LY0701	2007	Shandong/CN	EU022374.1	78.7%	87.6%	87.8%	81.2%	82.7%
39	CD0912	2021	Chengdu/CN	OK094646.1	79.1%	88.4%	87.8%	83.3%	82.8%
40	YB02	2021	Chengdu/CN	OK094645.1	79.1%	88.4%	87.5%	82.3%	82.7%
41	FJ1815	2019	Fujian/CN	MN052853.1	78.9%	88.0%	87.5%	81.6%	82.7%
42	YN180506	2019	Guangdong/CN	MK814585.1	78.4%	87.2%	87.6%	82.3%	82.4%
43	PT07	2008	Fujian/CN	EU499310.1	79.1%	88.4%	87.5%	80.9%	82.8%
44	AHAU28	2020	Anhui/CN	MT646348.1	78.8%	88.0%	87.6%	81.6%	82.7%
45	GX1104	2012	Guangxi/CN	JX241046.1	78.9%	88.0%	87.5%	81.9%	82.6%
46	GX1105	2012	Guangxi/CN	JX241045.1	78.9%	88.0%	87.7%	82.3%	82.7%

**Table 6 animals-14-00006-t006:** Complete genome sequence details of isolated DuCV strains.

Strain	Nucleotide and Deduced Amino Acid Lengths of Region in Isolated DuCVs			Identities of Nucleotide and Deduced Amino Acid (%)							
	Cap nt (aa)	Rep nt (aa)	Full Genome (nt)	DB2022-1		DB2022-2		DB2022-3		DB2022-4	
				Cap nt (aa)	Rep nt (aa)	Cap nt (aa)	Rep nt (aa)	Cap nt (aa)	Rep nt (aa)	Cap nt (aa)	Rep nt (aa)
DB2022-1	774 (257)	879 (292)	1993	-	-						
DB2022-2	774 (257)	879 (292)	1993	99.7 (100)	100 (100)						
DB2022-3	774 (257)	879 (292)	1993	99.7 (100)	100 (100)	100 (100)	100 (100)				
DB2022-4	774 (257)	879 (292)	1994	99.7 (100)	100 (100)	100 (100)	100 (100)	100 (100)	100 (100)	-	-

**Table 7 animals-14-00006-t007:** Homology comparison of the *VP1* gene of the identified DHAV with reference strains.

No	Strain	Year	Region	Accession No.	Genotype	Identities of Full Genome Sequence (%)	
						nt	aa
1	VF-10	2012	Shandong/CN	KC191683.1	DHAV-3	98.2%	99.6%
2	YT-57	2012	Shandong/CN	KC191691.1	DHAV3	98.6%	100.0%
3	YA140902	2016	Sichuan/CN	KX523288.1	DHAV3	98.3%	100.0%
4	ZP	2014	Shandong/CN	KM267028.1	DHAV3	98.5%	100.0%
5	GY140703	2016	Sichuan/CN	KX523285.1	DHAV3	98.2%	99.6%
6	SD151227	2016	Sichuan/CN	KX523287.1	DHAV3	98.1%	100.0%
7	A/dk/CHN/SD08/2019	2020	Shandong/CN	MN912703.1	DHAV3	97.9%	99.6%
8	JS	2020	Heilongjiang/CN	MN164467.1	DHAV3	99.7%	100.0%
9	QL150713	2016	Sichuan/CN	KX523286.1	DHAV3	98.3%	100.0%
10	YA151006	2016	Sichuan/CN	KX523289.1	DHAV3	98.5%	100.0%
11	JN-12	2012	Shandong/CN	KC191689.1	DHAV3	97.2%	99.6%
12	C-LX	2009	Beijing/CN	FJ626667.1	DHAV3	97.4%	99.2%
13	VF-40	2012	Shandong/CN	KC191686.1	DHAV3	96.7%	98.3%
14	C-PSY	2009	Beijing/CN	FJ626670.1	DHAV3	96.8%	98.3%
15	C-YCZ	2009	Beijing/CN	FJ626672.1	DHAV3	96.5%	97.9%
16	YT-63	2012	Shandong/CN	KC191693.1	DHAV3	94.0%	95.4%
17	BM	2011	Vietnam	JF925119.1	DHAV3	95.8%	97.5%
18	C-YZC	2009	Beijing/CN	FJ626673.1	DHAV3	95.7%	97.1%
19	C-PJK	2012	Sichuan/CN	KC282430.1	DHAV3	94.0%	96.2%
20	HY	2014	Vietnam	KM361878.1	DHAV3	95.1%	97.5%
21	GD1	2007	Jiangsu/CN	EU289393.1	DHAV3	94.7%	96.7%
22	YT-BX	2012	Shandong/CN	KC191694.1	DHAV3	94.2%	95.4%
23	NT	2014	Vietnam	KM361883.1	DHAV3	94.2%	95.8%
24	NC	2011	Vietnam	JF925121.1	DHAV3	91.0%	92.1%
25	DN2	2014	Vietnam	KM361877.1	DHAV3	91.0%	92.1%
26	KHO2	2014	Vietnam	KM361880.1	DHAV3	90.6%	90.8%
27	LA1	2014	Vietnam	KM361881.1	DHAV3	90.6%	90.4%
28	SY1	2007	Zhejiang/CN	EF407857.1	DHAV1	70.0%	76.7%
29	ZI07-3	2007	Zhejiang/CN	EF502170.1	DHAV1	69.9%	75.8%
30	SY2	2007	Zhejiang/CN	EF407858.1	DHAV1	70.2%	76.7%
31	ZJ0812	2011	Shanghai/CN	JF437541.1	DHAV1	69.6%	75.8%
32	SY6	2007	Zhejiang/CN	EF407862.1	DHAV1	70.4%	76.7%
33	C-GA	2008	Beijing/CN	EU687759.1	DHAV1	70.2%	75.8%
34	C-TX	2008	Beijing/CN	EU687760.1	DHAV1	70.0%	75.8%
35	C-PHG	2009	Beijing/CN	FJ626668.1	DHAV1	70.4%	75.8%
36	C-QDY	2009	Beijing/CN	FJ626671.1	DHAV1	70.7%	75.8%
37	C-YXQ	2008	Beijing/CN	EU687762.1	DHAV1	70.6%	75.8%
38	JX140	2008	Beijing/CN	KT072750.1	DHAV1	71.1%	76.7%
39	GD06	2018	Fujian/CN	MH778934.1	DHAV1	71.0%%	76.7%
40	CZ141017	2016	Sichuan/CN	KX523281.1	DHAV1	70.4%	76.7%
41	DY140903	2016	Sichuan/CN	KX523282.1	DHAV1	71.0%	76.2%
42	C-CQ3	2008	Beijing/CN	EU687761.1	DHAV1	70.4%	76.7%
43	LY0801-P30	2014	Shandong/CN	KJ524552.1	DHAV1	69.0%	77.5%
44	MY	2009	Sichuan/CN	GU363950.1	DHAV1	71.3%	77.1%
45	LY1903	2019	Shandong/CN	MN319537.1	DHAV1	70.2%	75.4%
46	SQ1903	2019	Shandong/CN	MN319539.1	DHAV1	70.6%	75.8%
47	SY0812	2011	Shanghai/CN	JF437538.1	DHAV1	70.0%	75.0%
48	LY0801-CP20	2014	Shandong/CN	KJ606043.1	DHAV1	70.3%	75.4%
49	LY0801-CP30	2014	Shandong/CN	KJ606045.1	DHAV1	70.3%	75.4%

**Table 8 animals-14-00006-t008:** Homology comparison of the σC gene of the isolated ARV with reference strains.

No	Strain	Year	Region	Accession No.	Genotype	Whole-Genome Sequence Identity (%)	
						nt	aa
1	HB06	2008	Beijing/CN	EU526387.1	I	98.3%	96.3%
2	XJ-2011-0315	2013	Heilongjiang/CN	KC963045.1	I	98.5%	97.2%
3	JR1	2006	USA	EF122836.1	I	98.8%	97.9%
4	LN09-1	2014	Beijing/CN	KP288843.1	I	98.8%	97.9%
5	HB10-1	2014	Beijing/CN	KP288833.1	I	98.8%	97.6%
6	BJ10-1	2014	Beijing/CN	KP288867.1	I	98.3%	97.9%
7	S113	1995	Canada	L39002.1	I	98.2%	96.3%
8	601SI	1999	Taiwan/CN	AF204947.1	I	98.0%	96.3%
9	1733	1997	USA	AF004857.1	I	99.4%	98.8%
10	GuangxiR1	2012	Guangxi/CN	KC183744.1	I	99.3%	98.5%
11	GuangxiR2	2013	Guangxi/CN	KF741732.1	I	99.3%	98.5%
12	Fahey-Crawley	2006	Brazil	DQ868789.1	I	99.5%	99.1%
13	176	1999	Canada	AF218358.1	I	99.5%	99.1%
14	G-98	2006	Huhhot/CN	DQ643975.1	I	99.7%	99.7%
15	2012-0129	2013	Heilongjiang/CN	KC963052.1	I	99.7%	99.7%
16	ARV-HeB01	2016	Heilongjiang/CN	KX451231.1	I	99.7%	99.7%
17	ARV-XY01	2016	Heilongjiang/CN	KX451229.1	I	99.5%	99.1%
18	SD-2010-0085	2013	Heilongjiang/CN	KC963039.1	I	99.4%	98.8%
19	LN05	2016	Heilongjiang/CN	KX451225.1	I	99.7%	99.7%
20	C-98	2006	Huhhot/CN	EF057397.1	I	99.6%	99.4%
21	T-98	2006	Huhhot/CN	EF057398.1	I	99.5%	99.4%
22	ARV-LN01	2016	Heilongjiang/CN	KX451221.1	I	99.6%	99.4%
23	ARV-LN06	2016	Heilongjiang/CN	KX451226.1	I	99.6%	99.4%
24	B-98	2006	Huhhot/CN	DQ643974.1	I	99.5%	99.1%
25	VA-1	2008	India	EU681254.1	I	98.9%	97.2%
26	GEL12 98M	2001	Netherlands	AF354225.1	I	77.3%	77.6%
27	Reo/Broiler/YTLY/161024a	2018	Shandong/CN	MK189468.1	I	73.7%	73.9%
28	916	2000	Taiwan/CN	AF297214.1	II	61.2%	58.1%
29	ARV/Crow/Kagawa/P3/2012	2016	Japan	LC164026.1	II	61.0%	56.6%
30	GEL13a98M	2001	Netherlands	AF354226.1	II	51.3%	54.4%
31	NC/98	2006	USA	DQ995806.1	II	61.0%	55.7%
32	TARV-MN3	2013	USA	KF872234.1	II	53.3%	56.7%
33	TARV-ONEIL	2013	USA	KF872231.1	II	52.0%	56.2%
34	TARV-Crestview	2013	USA	KF872238.1	II	52.5%	55.5%
35	T1781	2013	Hungary	KC865792.1	III	59.7%	50.8%
36	SD18	2020	Shandong/CN	MT747423.1	III	59.3%	50.5%
37	42563-4	2006	USA	DQ872801.1	III	58.9%	54.7%
38	GEL13b98M	2001	Netherlands	AF354227.1	III	58.1%	53.5%
39	D6	2019	Canada	MN879650.1	IV	45.6%	48.9%
40	D12	2019	Canada	MN879710.1	IV	55.5%	49.8%
41	Reo/USA/Broiler/1057NY/18	2021	USA	MW854823.1	IV	56.2%	49.5%
42	AVS-B	2010	Hungary	FR694197.1	IV	54.8%	48.6%
43	K1600657	2019	USA	MK583337.1	IV	54.7%	49.5%
44	NLI12 96M	2001	Netherlands	AF354230.1	IV	54.1%	50.2%
45	918	2000	Taiwan/CN	AF297215.1	V	57.0%	48.0%
46	1017-1	2000	Taiwan/CN	AF297216.1	V	56.5%	48.3%
47	NGN20 7-1 2b	2021	Japan	LC604650.1	V	55.9%	47.0%
48	Reo/PA/Broiler/19981/13	2015	USA	KR856993.1	VI	55.5%	49.3%
49	Reo/PA/Broiler/03476/12	2015	USA	KP727784.1	VI	55.8%	48.0%
50	Reo/PA/Broiler/03200/12	2015	USA	KP727785.1	VI	55.8%	48.0%

**Table 9 animals-14-00006-t009:** Physicochemical properties analysis of DuCV-Cap, DHAV-3-VP1, and ARV-σC.

Sequence	pI	Instability Index	Aliphatic Index	GRAVY
DuCV-KC851817 (2013)	10.74	58.9	50.47	−0.748
DuCV-OQ658501.1 (2022)	10.61	60.94	50.08	−0.765
DHAV-3-FJ626673 (2009)	6.62	51.32	83.67	−0.149
DHAV-3-OQ658504.1 (2022)	7.29	52.39	82.04	−0.163
ARV-EF122836 (2006)	5.0	41.68	89.72	−0.057
ARV-OR046324.1 (2022)	4.81	36.12	89.11	−0.036

## Data Availability

The virus strains obtained in this study are available in GenBank under accession nos. OQ658500 for DB2022-1/Heilongjiang/2022, OQ658501 for DB2022-2/Heilongjiang/2022, OQ658502 for DB2022-3/Heilongjiang/2022, OQ658503 for DB2022-4/Heilongjiang/2022, QQ659504 for DHAV3/DB2022-5/Heilongjiang/2022, and OR046324 for DB2022-6/Heilongjiang/2022.

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
