# Peer review of "Molecular Detection and Genetic Characterization of Vertically Transmitted Viruses in Ducks"

_animals, 2023, doi:10.3390/ani14010006_

Round 1

Reviewer 1 Report

Comments and Suggestions for Authors

The study, titled “Molecular Detection and Genetic Characterization of Vertically Transmitted Viruses in Ducks,” extensively examines the distribution and genetic diversity of four vertically transmitted duck pathogens including DHBV, DuCV, DHAV-3, and ARV - in duck populations. Using PCR and RT-PCR, the study identifies DHBV as the most prevalent, highlights common co-infections, and explores strain relationships. The findings hold substantial implications, including insights into the evolutionary dynamics of these pathogens and potential benefits for future vaccination strategies, enhancing duck health and disease management.

The main question addressed by the research is to investigate the distribution and genetic variation of four vertically transmitted duck pathogens: duck hepatitis B virus (DHBV), duck circovirus (DuCV), duck hepatitis A virus 3 (DHAV-3), and avian reoviruses (ARV) in duck populations. The study used PCR and RT-PCR assays to conduct an epidemiological analysis, revealing the prevalence of each the studied virus in the duck population. The research also explored the occurrence of coinfection, indicating that ducks can be simultaneously infected with two, three, or four of these vertical transmission agents. Genetic analysis further revealed the specific genotypes of the identified strains for DuCV, DHAV-3, and ARV. The findings contribute to understanding the evolution of these viruses and may aid in selecting appropriate strains for vaccination.

The research on the distribution and genetic variation of vertically transmitted duck pathogens addresses a specific gap in the field and is both original and relevant.

This study is novel and contributes valuable insights to the respective subject area. Although previous studies have explored these viruses, the epidemiological characteristics and co-infection rates of the studied viruses in ducks have not been previously documented or reported.

In addition to recognizing the significance of the research, the article exhibits several significant concerns, alongside a few typographical and grammatical errors that must be addressed before the manuscript can be deemed suitable for publication. Some of the major queries are as follows:

> Why did the authors opt to collect specifically 271 swab samples from the target population rather than the required 384 samples, considering that Thrusfield's 2007 calculation recommends a minimum of 384 samples based on a 50% expected prevalence and a 95% confidence interval? The same question applies to why a particular number of samples from a specific region were collected? Did the authors apply the Proportional allocation method for this purpose? In that case, what was the total population in the target region?

> Materials and Methods section does not have any details regarding the statistical methods employed. After examining the abstract, results, and discussion sections, it becomes evident that the findings were not presented with reference to statistical analysis. Merely discussing the results in terms of percentage values, without accounting for statistical analysis, lacks scientific rigor and fails to make a meaningful contribution to the scientific community.

> Line No. 304 (Table-9): Please refrain from including tabular data in the discussion section, as tables and figures are intended for the results section.

The conclusions are consistent with the evidence and arguments presented and they address the main question posed.

The references are appropriate. However, it is recommended to include maximum references from the past five years.

Comments on the Quality of English Language

Moderate English language editing

Reviewer 2 Report

Comments and Suggestions for Authors

The study conducted by Xinrong Wang et al. provides valuable insights into the distribution and genetic variation of vertically transmitted duck viruses, namely DHBV, DuCV, DHAV-3, and ARV. The research, conducted in China, employed PCR and RT-PCR assays to evaluate the prevalence of these viruses in a duck population, shedding light on the complex landscape of vertically transmitted infections.

One of the key findings is the prevalence ranking of these viruses, with DHBV emerging as the most widespread threat, followed by DuCV, ARV, and DHAV-3. The identification of coinfection in duck samples underscores the complexity of viral transmission dynamics, emphasizing the need for a comprehensive understanding of these interactions.

The genetic analysis provides a deeper understanding of the strains involved. All identified DuCV strains were classified under genotype 1, and the close clustering of the DHAV-3 strain with previously identified Chinese strains highlights the regional dynamics of virus evolution. The ARV strain clustering under genotype 1 further adds to the nuanced picture of viral diversity among ducks.

The significance of these findings lies in their potential impact on vaccination strategies. With DHBV being the most prevalent virus, and considering the coinfection rates, the study underscores the importance of tailored vaccination approaches. The identification of specific genotypes within DuCV, DHAV-3, and ARV provides essential information for selecting suitable strains for vaccination, enhancing the efficacy of preventive measures in the duck breeding industry.

I would like to highlight a concern regarding the sequence analysis figures. They appear to be entirely illegible, and I recommend that improvements be made in this aspect for better clarity.

In conclusion, Wang et al.'s study significantly contributes to our understanding of the molecular aspects of vertically transmitted duck viruses. The findings not only enrich our knowledge of viral evolution but also offer practical implications for developing targeted vaccination strategies, ultimately aiding in the management and control of these infections in duck populations.

Reviewer 3 Report

Comments and Suggestions for Authors        The current study revealed the prevalence of DuCV, DHAV-3, DHBV and ARV in ducks. It also demonstrates the coexistence of these viruses in healthy adult ducks. Those information is useful for  Ducks industry.       Prior to accepting this work, this manuscript would need some substantial adaptations.       1 the samples just from one duck factory in a certain area, the viruses detect whether can illustrate the genetic characterization of these virus evolution?       2 some representative pictures of PCR results should provide in the article.       3 the number of  these virus detect positive is very large, but the sequece analysis number is very limited, for example, just one ARV σC gene obtained in your research.  

Round 2

Reviewer 1 Report

Comments and Suggestions for Authors

I appreciate the authors effort to improve the overall quality of the manuscript. Consequently, I endorse the acceptance of the manuscript.

Comments on the Quality of English Language

Minor editing of English language required